# Delayed Medical Care of Underserved Middle-Aged and Older African Americans with Chronic Disease during COVID-19 Pandemic

**DOI:** 10.3390/healthcare11040595

**Published:** 2023-02-16

**Authors:** Edward K. Adinkrah, Sharon Cobb, Mohsen Bazargan

**Affiliations:** 1Department of Family Medicine, Charles R. Drew University of Medicine and Science (CDU), 1731 E. 120th St., Los Angeles, CA 90059, USA; 2Mervyn M. Dymally School of Nursing (MMDSON), CDU, Los Angeles, CA 90059, USA; 3Department of Family Medicine, University of California Los Angeles (UCLA), Los Angeles, CA 90059, USA

**Keywords:** ethnic groups, COVID-19, health service utilization, disease management

## Abstract

**Background:** While African American middle-aged and older adults with chronic disease are particularly vulnerable during the COVID-19 pandemic, it is unknown which subgroups of this population may delay seeking care. The aim of this study was to examine demographic, socioeconomic, COVID-19-related, and health-related factors that correlate with delayed care in African American middle-aged and older adults with chronic disease. **Methods:** In this cross-sectional study, 150 African American middle-aged and older adults who had at least one chronic disease were recruited from faith-based organizations. We measured the following exploratory variables: demographic factors (age and gender), socioeconomic status (education), marital status, number of chronic diseases, depressive symptoms, financial strain, health literacy, COVID-19 vaccination history, COVID-19 diagnosis history, COVID-19 knowledge, and COVID-19 perceived threat. The outcome was delay in chronic disease care. **Results:** According to the Poisson log-linear regression, higher level of education, higher number of chronic diseases, and depressive symptoms were associated with a higher level of delayed care. Age, gender, COVID-19 vaccination history, COVID-19 diagnosis history, COVID-19 perceived threat, COVID-19 knowledge, financial strain, marital status, and health literacy were not correlated with delayed care. **Discussion:** Given that higher healthcare needs in terms of multiple chronic medical diseases and depressive symptomatology but not COVID-19-related constructs (i.e., vaccination history, diagnosis history, and perceived threat) were associated with delayed care, there is a need for programs and interventions that assist African American middle-aged and older adults with chronic disease to seek the care that they need. More research is needed to understand why educational attainment is associated with more delayed care of chronic disease in African American middle-aged and older adults with chronic illness.

## 1. Background

In the United States (US), African Americans have experienced a significantly higher burden of COVID-19 mortality and morbidity compared to other racial and ethnic groups [1,2,3,4,5,6,7,8,9]. According to analysis of early data across 28 states in the US, the relative risk for COVID-19 related mortality was 3.5 times higher for African Americans than Whites [10]. Similarly, COVID-19 outcomes are shown to be worse in US counties that are predominantly African American [3]. Predominantly African American counties had COVID-19 infection rates that were three times higher and death rates that were five times higher than counties that were predominately White [3].

One of the mechanisms contributing to higher COVID-19 morbidity and mortality among African Americans, relative to their White counterparts, was the higher prevalence of chronic medical diseases and conditions, including hypertension, cardiovascular disease, diabetes, and obesity [11,12,13]. Compared to Whites, African Americans were at an increased risk of severe COVID-19 [14,15]. Underlying chronic health diseases, such as cardiac and respiratory diseases, also demand a more aggressive management and treatment if COVID-19 is diagnosed [16]. Moreover, African American older adults who are socially isolated are at a greater risk for poorer COVID-19 outcomes, due to decreased anti-viral immune response associated with social isolation [17]. Combined with a wide range of social factors, such as adversity and financial distress, multiple comorbidities imposed a major risk to the African American community during the COVID-19 pandemic. In addition, the increased incidence of severe COVID-19 among African Americans is attributed to higher infection rates that suggest COVID-19 disparities result from increased vulnerability [14,15]. Thus, it is essential to study the mechanisms contributing to COVID-19 disparities among African American individuals and communities.

Multiple lines of data suggest that COVID-19 was particularly a threat to older adults with multiple comorbidities [18,19]. COVID-19 related hospitalization and death rates increase with age [20]. Undesired COVID outcomes, such as hospitalization, are highest among older adults with underlying diseases [20]. As all people, particularly older adults with comorbidities, were strongly encouraged to practice social distancing, we could expect a decrease in healthcare utilization and treatments required for the management of physical chronic diseases. Such social isolation, combined with other risk factors, such as depression, may have contributed to worsened disease management of patients with chronic diseases [18]. Risk reduction of COVID-19 and chronic disease management for African American older adults with multi-morbidity are highlighted as difficult challenges due to the high vulnerability of this group, a fact which warrants critical attention [3,21,22,23,24].

Three main mechanisms are proposed as causes of increased COVID-19 burden among African American middle-aged and older adults: (1) Health care access and quality, (2) excessive risk of exposure, and (3) weathering processes [25]. Although healthcare access/use is named as one of these mechanisms, not much is known about factors that may contribute to reduced healthcare use or interruption in care of African American middle-aged and older adults [6,9,20,26]. During the COVID-19 pandemic, patients and healthcare providers have appropriately canceled or postponed many elective surgery and outpatient visits [27]. However, less is known about risk factors of delayed care use in African American middle-aged and older adults with chronic disease. This is important because delay in care and poor access to specialized care and treatment during the pandemic was associated with an increase in severely uncontrolled chronic diseases, including diabetes, particularly in vulnerable populations [28]. While virtual care ultimately allowed patients with chronic disease to return to the same visit rate, it took a while for the healthcare system and patients to adjust to such change at the onset of the pandemic [29]. However, disparities in access and the use of telemedicine as an alternative have also been documented [30]. A large racial gap in the utilization of telemedicine services in the geriatric population has been well-described [30]. Compared to their White counterparts, African American older men were less likely to participate in telehealth [31]. As a result, addressing factors that caused a delay in or avoidance of accessing health care among African American middle-aged and older adults with multiple chronic diseases during the COVID-19 pandemic remains a major public health concern [32]. 

### Aims

In response to the increased need for studying healthcare delays among middle-aged and older African American adults with chronic diseases [33], this study examined the frequency and correlates of delayed healthcare utilization (i.e., cancellations and postponements), including primary care and specialized visits, prescribed medication use or refills, oral and dental care visits, emergency department utilization, and other healthcare use (lab/blood work) during the second phase of the COVID-19 pandemic (i.e., when vaccination was widely available) among a sample of under-resourced African American middle-aged and older adults residing in the South Los Angeles, California. We characterized the correlates for such delays among underserved African American older adults. 

## 2. Methods

### 2.1. Design and Setting

This cross-sectional study was conducted in faith-based organizations in Service Planning Area 6 (SPA 6) of Los Angeles County, which includes the cities of South Los Angeles, Compton, Watts, and Willowbrook. We actively engaged faith-based organizations/churches from under-resourced and medically underserved South Los Angeles communities within a 10-mile radius of the study site. Ten (10) churches with an estimated 90% predominantly African American population were recruited into the study. Home to over one million residents, SPA 6 of Los Angeles County has the highest rate of overall poverty in Los Angeles County. Close to one-third of SPA 6 residents live below the federal poverty levels and are disproportionately affected by health disparities compared to the rest of Los Angeles County [34]. One example is the hospitalization rate for heart failure, which is significantly higher for SPA 6 than for the rest of Los Angeles County (rate of 700/100,000 population for South LA versus 350/100,000 population for Los Angeles County) [34]. Similarly, in terms of healthcare utilization among older adults, only 52% and 54% of adults aged 65 years and older are vaccinated with the influenza and pneumococcal vaccination, which is the lowest rate among all eight (8) service planning areas in Los Angeles County [34].

Additionally, predominantly African American churches have been pivotal in engaging and empowering members and community citizens to make positive health changes and actions toward health disease prevention and management, such as hosting and participation in blood pressure screenings [35,36]. Historically, the African American church is a trusted social institution that has assisted in the dissemination of evidence-based findings and practices. Even though they are understudied, African American churches commonly hosts various health care centered events and projects in their churches that involve parishioners, in addition to teaching, counseling, and supporting the clergy leaders and the congregation for various health interventions.

### 2.2. Recruitment and Sampling

Non-random sampling was used for recruitment. A total of 150 individuals were enrolled. Eligibility was based on African American ethnicity, age 55 or more, and presence of at least one chronic condition. Data were collected between 2021 and 2022. This research is a part of a larger study designed to assess the impact of a community-based participatory, faith-based, multidisciplinary, theoretical-based intervention to reduce the risk of COVID-19 and enhance chronic care management among underserved African American older adults. The study protocol was approved by the IRB of the Charles R. Drew University of Medicine and Science (CDU), Los Angeles (IRB #: 1663247-1). All participants signed a written informed consent form before being enrolled to this study. Participants received financial compensation.

### 2.3. Measurement

#### 2.3.1. Demographics Characteristics

We used age, gender, educational attainment, and marital status as the study covariates. Educational attainment was operationalized as a continuous variable (number of years for school attendance). Higher scores indicated more years of education. We asked our participants whether they were married or lived with a partner, which was analyzed categorically as either married/lived with a partner or not married/do not live with a partner.

#### 2.3.2. COVID-Related Constructs

We asked participants about COVID-19 vaccination history, COVID-19 diagnosis history, COVID-19 perceived threat, and COVID-19 knowledge using multiple items borrowed from the PhenX COVID-19 library (https://www.phenxtoolkit.org/, accessed on 12 March 2021). The PhenX (consensus measures for Phenotypes and eXposures) Toolkit is a publicly available, web-based catalog of recommended, well-established measurement protocols of phenotypes and exposures [37,38].

#### 2.3.3. Financial Strain

This variable was measured using five items. Participants were asked, “In the past 12 months, how frequently were you unable to: (1) buy the amount of food your family should have, (2) buy the clothes you feel your family should have, (3) pay your rent or mortgage, (4) pay your monthly bills, and (5) make ends meet?” Items were on a 5-level response scale ranging from 1 (never) to 5 (always). A total “financial strain” score was calculated, with an average score of the five items ranging from 1 to 5. A high score was indicative of greater financial difficulty. These items are consistent with Pearlin’s list of low SES individuals’ chronic financial difficulties (Cronbach alpha = 0.92) [39].

#### 2.3.4. Healthcare Utilization

Participants were asked eight (8) questions pertaining to the use of specific medical and dental healthcare. For example, we asked: “Have you needed to see your primary care doctor/provider but you did not or delayed it because of the risk of contracting coronavirus?” Each question was followed by, “If yes, did you delay it? Or did you not see your doctor/provider at all?” These items include (1) prescription medication refill(s), (2) primary care visits, (3) specialty care providers, such as cardiologist, etc., (4) dentist, (5) surgical procedures, (6) emergency department/room, (7) physical therapy, lab work, x-ray, etc., and (8) mental health professionals. We calculated the total number of healthcare utilization delays, which were defined as cancellations or postponements of healthcare visits.

#### 2.3.5. Number of Chronic Diseases

We asked participants to report whether they have been diagnosed with the following diseases: high blood pressure/hypertension, diabetes mellitus, cardiovascular disease, chronic kidney disease, cancer, stroke, chronic obstructive pulmonary disease (COPD), asthma, HIV, hepatitis B virus (HBV), hepatitis C virus (HCV), tuberculosis (TB), alcohol or substance use disorder, low back pain, migraine headache, and any other physical chronic condition. This variable had a potential range of 0 to 16 but an actual range of 1 to 6, with a higher number indicating multi-morbidity.

#### 2.3.6. Subjective Health

Knowing that single item self-rated health (SRH) measure is differently shaped by social determinants across ethnic groups [40], this study measured subjective health by three questions. Participants were asked to rate their physical health, mental health, and oral health with the options of (a) excellent, (b) very good, (c) good, (d) fair, and (e) poor. We calculated the average of these three items as our subjective health with higher score indicating worse subjective health. These SRHs are shown to be valid predictors of morbidity and mortality [41].

#### 2.3.7. Depressive Symptoms

This study used the 14 items to evaluate depression, asking participants “In the past 7 days, including today, how often were you distressed by: (1) Feeling no interest in things; (2) Nervousness or shakiness inside, (3) Feeling lonely, (4) Feeling tense or keyed up, (5) Nausea or upset stomach, (6) Feeling blue, (7) Suddenly scared for no reason, (8) Feeling hopeless about the future, (9) Feeling fearful, (10) Feeling super alert or watchful on guard, (11) Having difficulty concentrating, (12) Trouble experiencing positive feelings, (13) Feeling guilty or blaming yourself, and (14) Feeling irritable, angry, or aggressive.” Responses were on a five-point scale: “not at all = 1” to “all the time = 5”. A summary score was calculated by taking a mean of all items with a potential range between 1 and 5, in which a higher score indicated more depressive symptoms. 

#### 2.3.8. Health Literacy

We used four (4) items to measure health literacy. We asked participants to report (1) “How often do you have someone help you read hospital materials?”; (2) “How often are you confident when filling out medical forms by yourself?”; (3) “How often do you have problems learning about your medical condition because of difficulty understanding written information?”; and (4) “How often do you have a problem understanding what is told to you about your medical condition?” Responses were on a five-point scale: “not at all = 1” to “always = 5”. A summary score was calculated by taking a mean of all items with a potential range between 1 and 5, in which a higher score indicated more difficulty. 

### 2.4. Data Analysis

Our analysis had three parts. The first section was a descriptive analysis of all participants. This descriptive work reported means and standard deviations for continuous measures and frequencies and percentages for the categorical variables. Next, we conducted Pearson correlation to examine the bivariate association between all study variables including socio-demographics, COVID-19-related measures, health, and outcome variable (delay in healthcare utilization). For multi-variable analysis, we used a generalized linear model with Poisson log-linear to examine the independent association of delayed of healthcare utilization and all our exploratory variables. Both categorical and continuous independent variables were included in the model regardless of the results of bivariate analysis. For Poisson log-linear regression, the exponentiation of the B coefficient (Exp (B)), which is an odds ratio as well as 95% confidence interval are reported. For multivariate analysis, *p* values of less than 0.05 were considered significant. 

## 3. Results

### 3.1. Descriptive Analysis

Table 1 reports the descriptive results of the 150 African American older adult participants. All participants were between the ages of 55 and 91 years (mean= 68.5 ± 8.66). Approximately 23% of participants were 75 years of age or older, with only 39% self-reporting being married. Thirteen percent of participants never completed high school and another 27% reported completing high school. The main domain in which our participants reported financial distress was paying medical bills. More than 79% of participants indicated that they always (72%) or often (8%) have difficulty paying their medical bills. Only thirty percent (30%) of participants self-reported their physical health as very good or excellent. Almost 28% reported their oral health as poor or fair. 

Almost one out of three (32%) participants indicated that they delayed at least one type of medical care due to the COVID-19 pandemic. More than 18% and 6.8% and 6.7% reported one, two, or at least three types of medical services that they needed but delayed, respectively.

### 3.2. Bivariate Analysis 

Table 2 shows bivariate correlations between delayed care and other study variables. These correlations are calculated based on Pearson correlation test. As this table depicts, age and gender were not correlated with delayed care. However, level of education, the number of chronic diseases, and depressive symptoms were all associated with delayed care. However, COVID-19 vaccination history, COVID-19 diagnosis history, COVID-19 perceived threat, COVID-19 knowledge, financial strain, marital status, and health literacy were not correlated with delayed care.

### 3.3. Multi-Variable Analysis

Table 3 presents the summary of multi-variable analysis using Poisson log-linear regression. According to this table, education was negatively associated with delayed care, whereas the number of chronic diseases and depressive symptoms were positively associated with delayed care. Age, gender, COVID-19 vaccination history, COVID-19 knowledge, COVID-19 diagnosis history, COVID-19 perceived threat, financial strain, marital status, and health literacy were not correlated with delayed care.

## 4. Discussion

We conducted this study to examine frequency and correlates of delayed healthcare utilization (cancellations and postponements) during the second phase of the COVID-19 pandemic (when vaccination was widely available) among a sample of underserved African American older adults residing in the SPA 6 region of California. We characterized three correlates of delayed healthcare use among underserved African American older adults with chronic disease: higher education, higher number of chronic diseases, and depressive symptoms.

We observed a positive association between a higher number of chronic diseases and delayed care, which reflects multiple and competing needs. Individuals with multiple chronic diseases may have to seek specialized, higher levels of care, and result in greater needs and a higher likelihood of a delay. The theory of scarcity explains that in the presence of multiple demands and short supply, individuals may not be able to actively and efficiently allocate their resources and may feel overwhelmed, such as our study population. Several studies that examined the impact of delayed medical care on health outcomes revealed that patients with chronic conditions who delayed medical care, particularly older adults, may be at risk for subsequent hospital admissions and premature death [42,43,44]. In addition, evidence shows that delayed use of healthcare among older adults is negatively associated with both self-reported health and mental health status [45]. One recent study shows that one in five older adults who experienced delayed medical care during the COVID-19 pandemic reported that their health conditions were negatively affected by the care delay [46].

As a predictor of delayed care, depressive symptoms is shown to reduce motivation and limit achievement of goals, especially when they are complex and require multiple steps, such as making an appointment with a healthcare provider [47]. Depression has also been shown to cause delay in healthcare use and decrease interest in attaining positive healthcare outcomes [48]. However, very few studies have tested the effects of depression on disease management among African American older adults during the COVID-19 pandemic, particularly for ethnic differences in depressive symptomatology and as a comorbidity with other chronic diseases [49].

Regarding the role of higher education as a predictor of delayed care, we propose four potential explanations. First, higher education may be a proxy for culturally based social and family roles among African Americans. A study by Thorpe a showed higher number of roles (work–life imbalance) is a risk factor of obesity in African American men [50,51]. Our second explanation is the diminished returns of education in African American communities. In several studies, education is shown to be a risk factor for depression and other risk factors in African American communities. This may be attributed to higher socioeconomic statuses among African Americans, who may experience social isolation when residing in wealthier communities. Due to their upward social mobility, they likely moved from their former neighborhood, which may have been more aligned with their cultural and spiritual values and beliefs. This explanation may hold true, as some participants may have moved from other rural and lower socioeconomic areas (US Southern states) to South Los Angeles. Our third explanation is a statistical one. As we had a small non-random sample, our results are not representative of all African Americans with chronic disease, and a larger random sample may be the next step for research. Related to this explanation, this finding may be specific to this region, which is low income and historically underserved. Finally, due to structural racism, we observe weaker than expected effects of education on the promotion of living conditions of African American older adults. This is particularly the case given the age of our participants. Some of our participants attended highly segregated schools. A study conducted in South Los Angeles found that financial education was correlated with poor health outcomes across multiple domains, yet educational attainment failed to protect against any conditions. This is in part because of structural factors, such as social stratification and segregation and chronic poverty. Under chronic poverty, education may not be enough to secure health outcomes for the historically oppressed. There is a need for systemic change so that chronic poverty as the root cause can be addressed and solutions provided [52,53,54].

The findings of our study have major implications because it was conducted among African American older adults, a group who experienced a higher burden of COVID-19 compared to the general population [1,2,3,4,5,6,7,8,9]. Our data regarding poor healthcare use and interrupted disease management may partially explain why the relative risk for COVID-19 mortality was higher for African Americans than Whites in the US [10]. Although many other structural factors are also involved, our findings may help clinicians and policymakers understand the worse COVID-19 outcomes in African American communities [3]. We argue that poor disease management of patients with multi-morbidity may reduce resilience and increase the susceptibility of African Americans to COVID-19, as hypertension, cardiovascular disease, diabetes, and obesity are all risk factors for poor COVID-19 outcomes [11,12,13]. In addition, the increased incidence of severe COVID-19 among African Americans is also due to higher infection rates suggesting that COVID-19 disparities most likely result from societal and historical processes, such as segregation and social stratification [14,15].

Our study is important regarding multiple disadvantages that place low-income underserved African American older adults with multiple chronic diseases at risk of COVID-19 burden [18,19]. Through poor management of underlying diseases, African American older adults with chronic disease are at a higher risk of hospitalization during COVID-19, as shown before [20]. So, while populations are strongly encouraged to practice social distancing and isolation as an effective prevention measure, this should not result in delays in health management and care of comorbidities in African American older adults. Therefore, policymakers and clinicians should seek solutions to educate populations to avoid a delay in healthcare utilization and treatments that are essential and required for management of their physical and mental chronic diseases. This is potentially possible through community-based interventions that maximize vaccination uptake, correct adherence to masking, and other preventive measures, while enabling the populations to get their required care. Provision of telehealth and telemedicine can serve as effective solutions to address the challenge of care provision at the time of pandemics.

This study also propels policymakers and clinicians to consider that social isolation can also be a side effect of avoidant healthcare practices, including delay of healthcare. Such delays can themselves lead to other adverse outcomes, such as poor management of chronic diseases [18]. Recent data published by Centers of Disease Control and Prevention (CDC) shows that since late 2020 and early 2021, the rate ratios of COVID-19 incidence, emergency department visits, hospital admissions, and deaths among older adults have declined significantly due to availability and prioritization of COVID-19 vaccination among this segment of our population [55].

Several studies clearly documented that African American older Adults with comorbidities are most vulnerable to COVID-19 infection and face the most severe and critical consequences of this pandemic [6,9,20,26]. As stated previously, three primary factors that exacerbated the COVID-19 burden among African American older adults are: (1) excessive risk of exposure, (2) healthcare access and quality, and (3) weathering processes [25]. In addition to older age, underlying health diseases, including cardiac and respiratory diseases, will lead to a more severe and aggressive management of COVID-19 [16]. Moreover, African American older adults practicing social isolation are at greater risk for poorer outcomes if exposed to COVID-19 due to a decreased anti-viral immune response associated with social isolation [17]. Risk reduction of COVID-19 and chronic disease management for African American older adults with multi-morbidity are highlighted as difficult challenges due to the high vulnerability of this group, a fact which warrants critical attention [3,21,22,23,24].

It is well-established that during the COVID-19 pandemic, patients and healthcare providers have appropriately canceled or postponed both many elective surgery and outpatient visits [27]. Indeed, access to specialized care, such as diabetes and hypertension services, declined sharply during the pandemic among populations with severely uncontrolled diseases [28]. The decrease of in-person care during the pandemic was accompanied by an increase in virtual care, which ultimately allowed patients with chronic disease to return to the same visit rate as they had before the onset of the pandemic [29]. However, early signs of disparities in access to care delivered through telemedicine have been documented [30]. A large racial as well as gender gap in the utilization of telemedicine services in the geriatric population have also been documented. African American men, particularly older men, are less likely to participate in telehealth than their White counterparts [31]. Adverse short- and long-term outcomes caused by delay or avoidance of accessing healthcare among older adults with multiple chronic diseases during the COVID-19 pandemic continue to be major public health concerns [32]. 

It is noteworthy that 30% of our African American older adults who participated in this study are living alone and may not have reliable transportation and assistance for healthcare appointments, particularly due to the pandemic. Therefore, it is important to further examine the combined impact of household composition and accessibility to appropriate transportation. It is well documented that transportation barriers to health care have a disproportionate impact on underserved individuals who have chronic conditions [56]. Data from the National Health Interview Survey shows that the risk of delayed care due to cost or lack of transportation is particularly high for underserved older adults living alone [57]. Indeed, older adults who are transportation-disadvantaged experience limited access to health care, goods and services, and are isolated from familiar lifestyle habits and social networks [58]. Previous studies suggest that while non-adherence to medicated and non-medicated treatment plans remains an important issue among older adults, for underserved older African Americans, the non-adherence is particularly striking and requires urgent interventions [59]. Taken together, social isolation and additional psychological impacts of the pandemic underscore the importance of interventional efforts to older adults [60]. Promoting social connection, mobilizing the resources from both family members and community-based networks (such as faith-based organization, etc.), and engaging the healthcare system to identify underserved older adults who are living alone with transportation disadvantages are a few strategies, particularly during pandemic, that should be considered in healthcare settings [61].

### 4.1. Future Research

Further research on healthcare delays during the pandemic, particularly among older adults with comorbidities, is needed [33] to design appropriate intervention strategies to address these racial disparities in healthcare delivery. Additional studies should investigate these findings in other contexts using randomized samples. Future research may explore how each chronic disease contributes to delay in health care for other racial and ethnic populations, such as Latino and Asian/Pacific Islander older adults in South Los Angeles. Finally, and most importantly, researchers should seek ways to dismantle these inequities. Structural factors, policies, and the characteristics of neighborhood, insurer, and healthcare system may explain some of the variance of delay in healthcare use. Researchers may also test the role of stigma, discrimination, and other social identities, such as cultural beliefs, which can contribute to these findings. Larger random national data sets should be used to test whether these findings are specific to South Los Angeles or if similar patterns can be seen across the US.

### 4.2. Limitations

This study had several limitations. First, this study had a small sample size, which limits our confidence regarding the generalizability of findings, particularly negative associations. That is, we cannot claim that associations do not exist. Instead, we report that we did not find any association in our sample. In particular, this limitation could explain the lack of association between COVID-19 vaccination status and delay in care in this study. Similarly, we expected financial distress to increase delay in health care, which was not observed, potentially because of the low sample size. A second limitation was the study’s cross-sectional design. All our study variables were measured at an individual level as we did not collect neighborhood level data or data from the healthcare system [35,36].

As discussed previously, this study was a community-faith-based survey, and we had no access to their medical charts or information. Participants were using different sites for receiving their medical care; therefore, we are not able to document, for example, each clinic’s no-show rate or cancelation policy. Moreover, we did not assess employment status as our study population focused on older adults who may be more likely to be retired/unemployed. We also did not differentiate between different chronic diseases and their severity. While we pooled all chronic diseases together, it is possible that some conditions may differently correlate with delay in healthcare use. Finally, we did not measure the time window for a visit cancelation (e.g., number of days prior). 

## 5. Conclusions

To conclude, one out of five who delayed their care never received the care, even one year after the COVID-19 vaccination was widely available. This rate is concerning given the established literature on the role of continuity of care for the management of chronic disease. The findings may partially explain the high vulnerability of African American middle-aged and older adults with chronic disease to COVID-19 morbidity and mortality. Our findings also suggested that African American middle-aged and older adults with high depressive symptoms and those with higher number of chronic diseases are at a higher risk of delaying their required care.

## Figures and Tables

**Table 1 healthcare-11-00595-t001:** Demographic, Financial, and Health-related Characteristics of Sample (n = 150).

	N (%)		
Gender Male Female	45 (30) 105 (70)		
Age 55–64 65–74 75 and older	48 (32) 68 (45) 34 (23)		
Education No High School Diploma High School Diploma Some College/Graduate	19 (13) 40 (27) 91 (60)		
Married/Partner No Yes	92 (61) 58 (39)		
Delayed care None One visit Two visits Three or Four Visits	101 (68.2) 27 (18.2) 10 (6.8) 10 (6.7)		
	**Mean ± SD**		
Number of Chronic Diseases (0–6)	1.91 ± 1.43		
Depressive Symptoms	1.38 ± 0.62		
**Quartiles**	**25%**	**50%**	**75%**
Financial Strains (1 = never to 5 = always)	1.60	1.80	2.25

**Table 2 healthcare-11-00595-t002:** Bivariate Correlations Between Delayed Care and Other Study Variables.

	1	2	3	4	5	6	7	8	9	10	11	12
Delay of Care	1											
2. Gender (Male)	−0.043	1										
3. Age	−0.119	−0.071	1									
4. Education	0.172 *	−0.017	−0.143	1								
5. Married/Companion	0.011	0.227 **	−0.192 *	0.029	1							
6. Financial Strain	0.084	0.056	−0.135	−0.237 **	−0.077	1						
7. Health Literacy	−0.102	0.023	0.213 **	−0.386 **	0.040	0.135	1					
8. COVID-19 Knowledge	0.069	−0.060	−0.041	0.115	−0.004	−0.180 *	−0.030	1				
9. COVID-19 Vaccination	0.075	−0.141	0.110	0.095	0.025	−0.114	−0.242 **	0.026	1			
10. COVID-19 Diagnosis	0.053	−0.029	−0.031	0.094	−0.002	0.031	0.002	−0.029	0.171 *	1		
11. Chronic Diseases (n)	0.234 **	−0.050	0.046	−0.109	0.052	0.190 *	0.072	0.137	0.146	0.110	1	
12. COVID-19 Perceived Threat	0.094	−0.090	0.048	−0.021	−0.075	0.045	0.124	−0.012	−0.018	0.195 *	0.196 *	1
13. Depression Symptoms	0.230 **	−0.045	−0.081	−0.102	−0.005	0.366 **	0.218 **	−0.086	−0.034	0.024	0.260 **	0.262 **

Notes: * indicates that Correlation is significant at the 0.05 level (2-tailed). **—Correlation is significant at the 0.01 level (2-tailed).

**Table 3 healthcare-11-00595-t003:** Poisson Log-Linear Regression, the Odds Ratio (OR) and 95% Confidence Interval).

Independent Variables	Wald Chi-Square	OR	95% Confidence Interval		Sig
			Lower	Upper	
Gender: Male	0.024	1.044	0.607	1.795	0.877
Age	1.697	0.978	0.946	1.011	0.193
Education	6.211	1.339	1.064	1.685	0.013
Marital Status: Married/Partner	1.014	1.301	0.780	2.170	0.314
Financial Strain (Low to High)	0.431	1.117	0.803	1.554	0.511
Health Literacy: (High to Low)	0.152	0.933	0.657	1.324	0.696
Knowledge of COVID	2.194	1.999	0.799	5.002	0.139
Vaccine: Completed	0.774	0.722	0.350	1.491	0.379
Diagnosed with COVID: Self or Family	0.544	1.209	0.730	2.002	0.461
Number of Chronic diseases	4.749	1.199	1.018	1.412	0.029
Perceived Thereat COVID	0.209	1.028	0.914	1.156	0.647
Depression symptoms	6.106	1.517	1.090	2.112	0.013
Intercept	2.329	0.075	0.003	2.090	0.127

## Data Availability

The data sets used and analyzed in the current study are available from the corresponding author for collaborative studies.

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
