# Peer review of "Delayed Medical Care of Underserved Middle-Aged and Older African Americans with Chronic Disease during COVID-19 Pandemic"

_healthcare, 2023, doi:10.3390/healthcare11040595_

Round 1
Reviewer 1 Report
Summary:
The submitted manuscript evaluates the impact of COVID-19 pandemic on decreased healthcare utilization among Low-Socioeconomic Middle-aged and older Black adults with Chronic Illness. The authors incorporate demographic and general health related variables to model their impact on the number of appointment cancellations or postponements.
Comments:
Line 61: What is CMD?
Table 1:
- Financial Strains and Self related health can be better described by Median and Inter Quartile range.
- Chronic illnesses: Charlson or Elixhauser comorbidity index are better suited to measure this variable.
Line 108: "This cross-sectional study was conducted in faith-based organizations in South Los Angeles". Could the authors expand on faith-based organizations, and their connection to the subjects. It is is important to understand the resources available at these organizations and their role in a subjects healthcare utilization.
The study subjects were recruited from SPA6. What is the demographic breakdown of this service planning area. Are the selected subjects under-represented in this area?
If the subjects are indicative the general socio-economic makeup of this area, then their problems can be considered as endemic to other service area with similar socio-economic makeup.
Since the study is structured as a comparative analysis, is there a baseline year for us to compare and show that COVID related reasons have increased cancellations/postponements?
Outcome variable: The authors define delayed care as the number of cancellations and postponements. I have several questions about this definition:
- Why is this not described in Table1. We need to understand the distribution. Are there outliers(i.e., people with many cancellations that might bias the model)
- Is there a time window for cancellations that make it a true cancellation? If subjects cancelled/postponed by a couple of days, would it still be treated as a true cancellation.
- Cancellations and postponements can not be treated the same. Postponements could mean the subjects still received care in a timely manner, while cancellations would be true indicators of denied healthcare access.
- Each health system usually has a specific no-show rate. Were the cancellations/postponements higher during the pandemic for this health system?
- Prescription refills are included as part of the health care utilization metric. I do not know how this works at this study site, but do patients generally need an appointment to get a prescription refill?
- Can we get a breakdown of cancellations/postponements for the different categories listed under healthcare utilization?
Line 120: Education was a continuous variable. Why not use standard definitions from table 1(No High School Diploma, High School Diploma, Some college/graduate) instead of years, as this can be misleading.
For example 11 years can be freshman/sophomore dropout, while 13 years can indicate high school diploma. In this scenario, the additional 2 years can indicate 2 completely separate populations.
The authors provided multiple explanations for how education can be positively corelated with cancellations in Line 245-267. If job responsibilities could be an indicator for cancellations, why not use current employment status as a variable?
General questions:
What role does health insurance play in this study?
What role does transportation play in this study? I ask this because ~30% of the subjects are living alone, and they might not have the help available to get them to appointments.
Did these healthcare orgs have telehealth implemented? And if they did, were these services available to the subjects?
Suggestions:
- I would like the authors to better describe their study setting. There are too many gaps for the average reader to fill in.
- The outcome variable needs to be elaborated upon. Providing data distributions, and breakdown across the various categories is necessary.
- To show cause and effect, the authors need to show what effect the missed appointments had on a subjects health. This could be defined as increase in mortality, unplanned emergency department usage, longer consequent hospital stays or some adverse event due to the lack of access to timely care.
Reviewer 2 Report
In the discussion, results from previous researches should be added, specifically researches on minority races of other countries and whether similar things had happened.

Author Response
Thank you - Please see attached file

Round 2
Reviewer 1 Report
I am happy with the changes and justification provided by the authors to my earlier review. I propose to accept this manuscript.
Reviewer 2 Report
accepted